# Removal of lycopene substrate inhibition enables high carotenoid productivity in *Yarrowia lipolytica*

Yongshuo Ma [1,2], Nian Liu[1,3], Per Greisen[4], Jingbo Li [1], Kangjian Qiao[1], Sanwen Huang [2✉] & Gregory Stephanopoulos [1✉]

Substrate inhibition of enzymes can be a major obstacle to the production of valuable chemicals in engineered microorganisms. Here, we show substrate inhibition of lycopene cyclase as the main limitation in carotenoid biosynthesis in *Yarrowia lipolytica*. To overcome this bottleneck, we exploit two independent approaches. Structure-guided protein engineering yields a variant, Y27R, characterized by complete loss of substrate inhibition without reduction of enzymatic activity. Alternatively, establishing a geranylgeranyl pyrophosphate synthase-mediated flux flow restrictor also prevents the onset of substrate inhibition by diverting metabolic flux away from the inhibitory metabolite while maintaining sufficient flux towards product formation. Both approaches result in high levels of near-exclusive β-carotene production. Ultimately, we construct strains capable of producing 39.5 g/L β-carotene at a productivity of 0.165 g/L/h in bioreactor fermentations (a 1441-fold improvement over the initial strain). Our findings provide effective approaches for removing substrate inhibition in engineering pathways for efficient synthesis of natural products.

[1] Department of Chemical Engineering, Massachusetts Institute of Technology, Cambridge, MA 02142, USA. [2] Shenzhen Branch, Guangdong Laboratory of Lingnan Modern Agriculture, Genome Analysis Laboratory of the Ministry of Agriculture and Rural Affairs, Agricultural Genomics Institute at Shenzhen, Chinese Academy of Agricultural Sciences, Shenzhen 518120, China. [3] Bristol Myers Squibb, 400 Dexter Ave N, Seattle, WA 98109, USA. [4] Global Research, Novo Nordisk A/S, Måløv DK-2760, Denmark. ✉email: huangsanwen@caas.cn; gregstep@mit.edu

Engineering microbes for the production of valuable chemical products is an attractive alternative to sourcing these compounds from nature or deriving them from petrochemicals by chemical synthesis[1–3]. However, synthetic biology efforts to achieve economically viable and scalable titers and productivities are frequently hindered by undesirable regulatory mechanisms that modulate the activity of enzymes. Such mechanisms have evolved to mediate optimal cellular response to changing physiological conditions, but also represent a major obstacle in redirecting metabolic fluxes toward desired engineered metabolic pathways and away from native growth-optimizing ones. This problem is particularly evident in compounds that require long and complex synthesis pathways (e.g., isoprenoids), frequently giving rise to bottlenecks that may reduce cell fitness and pathway productivity[4,5]. As such, it is imperative to develop methods that allow us to circumvent the effect of enzyme inhibition in constructing robust strains with high productivity.

Substrate inhibition represents one such enzyme-level regulation deployed in cells to help optimize cellular economy and maximize growth in response to temporal variations of the environment[6]. Moreover, such mechanism is often used to design therapies for various diseases[7,8]. However, it is undesirable in industrial applications of microbes mediated by enzymatic reactions for product synthesis. Enzyme inhibition is typically triggered when substrate concentration exceeds a certain threshold, thus preventing the catalytic conversion of the substrate and limiting the flux through the desired pathway. As such, substrate inhibition is particularly detrimental to the synthesis of end-products of interest when present in the middle of a metabolic pathway, which in turn causes intermediates accumulation, pathway disruption and alteration in the profile of products formed. Although several methods have been explored to address this limitation, such as enzyme immobilization[9,10], two-phase partitioning bioreactor systems[11–13], batch substrate-feeding strategy[14], and protein engineering[15,16], most of these solutions are limited to systems where the inhibition is posed by the starting substrate, and difficult to apply in the context of microbial engineering for chemical production.

In this work, using the oleaginous yeast Yarrowia lipolytica for carotenoid overproduction, we demonstrate two independent strategies that nearly completely circumvent substrate inhibition. First, the enzyme lycopene cyclase is identified as the bottleneck in the synthesis of carotenoids due to its strong substrate inhibition by lycopene. This results in not only low titers of β-carotene but also large amounts of accompanying lycopene as byproduct. In light of this, our first strategy is to use a structure-guided protein design, coupled with phylogenetic information, to generate protein variants with reduced inhibition. Of the 50 variants generated, a single mutation Y27R is identified that completely abolished substrate inhibition without reducing enzyme activity, resulting in a remarkable increase of β-carotene production and 98% selectivity (% product vs. sum of all carotenoids). Alternatively, in the second approach, similar titers and selectivity of β-carotene are obtained by reducing the carbon flow through the carotenoid pathway and thus preventing inhibitory metabolite accumulation to inhibitory levels, contrary to the traditional paradigms of pathway engineering. This is achieved by establishing a geranylgeranyl pyrophosphate synthase (GGPPS)-mediated metabolic flow restrictor that regulates the substrate lycopene formation rate, thereby effectively alleviating substrate inhibition. While this approach reduces flux through the pathway of interest, the gains from suppressing substrate levels and thus maintaining high enzymatic activity overcompensate for any losses in productivity suffered from the flux diversion. Using the methods outlined above, along with careful partitioning of cellular resources dedicated to carotenoid synthesis versus storage,

we ultimately establish a strain capable of producing 39.5 g/L β-carotene (98% selectivity) with a 0.165 g/L/h volumetric productivity in bioreactor fermentations. Moreover, by deliberately exploiting the substrate inhibition effect, we are also able to shift the product profile towards lycopene instead, achieving lycopene titers of 17.6 g/L and productivities of 0.073 g/L/h. Overall, our findings highlight the importance of and provide methods for abolishing substrate inhibition in engineering cell factories for biotechnological production of high-value compounds.

## Results

**Substrate inhibition of lycopene cyclase limits carotenoid synthesis.** Synthesis of β-carotene in Y. lipolytica requires heterologous expression of three genes encoding the enzymes phytoene synthase, phytoene dehydrogenase, and lycopene cyclase (Fig. 1a and Supplementary Fig. 1). Additionally, geranylgeranyl diphosphate synthase (GGPPS) should also be considered as it controls the flux directed towards carotenoid instead of sterol synthesis (Fig. 1a and Supplementary Fig. 1). We sourced the relevant genes from the eukaryotic organisms, Xanthophyllomyces dendrorhous and Mucor circinelloides, for expression. Since Y. lipolytica already harbors a native copy of GGPPS, we started by introducing gene expression cassettes encoding phytoene dehydrogenase and the bi-functional phytoene synthase/lycopene cyclase from X. dendrorhous (CrtI and CrtYB, respectively)[17,18] or M. circinelloides (CarB and CarRP, respectively)[19,20] into the Y. lipolytica po1f strain with TRP1 disruption (po1f-T) (Supplementary Fig. 2 and Table 1). Strain YLMA02, which expressed enzymes from M. circinelloides, produced 4.12-fold more β-carotene (27.4 mg/L) than strain YLMA01, which expressed enzymes from X. dendrorhous (Fig. 1b). Thus, the CarB/CarRP pair was used in all further studies.

Although synthesis of β-carotene was observed in YLMA02, the titers were very low, prompting us to investigate the GGPPS step as the next target. Introduction of an additional copy of GGPPS from X. dendrorhous (GGPPxd) into strain YLMA02 significantly increased the titers of β-carotene to 0.48 g/L (Fig. 1c). However, this was also accompanied by a large increase in lycopene accumulation (Fig. 1c), which suggested that the cyclization from lycopene to β-carotene was a pathway bottleneck. Correspondingly, we attempted to enhance the expression of lycopene cyclase by increasing its gene copy number. Moreover, since cyclase activity is conferred by the R domain of the bifunctional enzyme CarRP[20], we also introduced a modified version of the protein with its P domain either deleted or mutated (Supplementary Fig. 3a), such that it served as a dedicated cyclase. None of these efforts succeeded in improving β-carotene titers (Supplementary Fig. 3b), despite the higher mRNA level was observed (Supplementary Fig. 3c). Next, we examined the effect of introducing lycopene cyclases (EuCrtY, PaCrtY, PfCrtY and HpCrtY) from four other organisms (Supplementary Table 1), which led to modest improvements in β-carotene production (Supplementary Fig. 3d). However, the issue of lycopene accumulation remained largely unaddressed, suggesting that the overexpression of lycopene cyclase, regardless of its origin, was not an effective strategy. It was very possibly that the protein level of lycopene cyclase was not the limitation in this study.

Since lycopene was the only aggregating precursor (Supplementary Fig. 4) and adding additional copies of various lycopene cyclases did not circumvent the issue, we hypothesized that the activity of lycopene cyclase was inhibited by excess lycopene through substrate inhibition (Fig. 1a). To test this hypothesis, we examined possible correlations between lycopene cyclase activity and lycopene concentration using an in vitro enzymatic assay that employed a yeast microsomal system, as CarRP is predicted to be a membrane protein with six transmembrane helices (Supplementary Fig. 5). The results

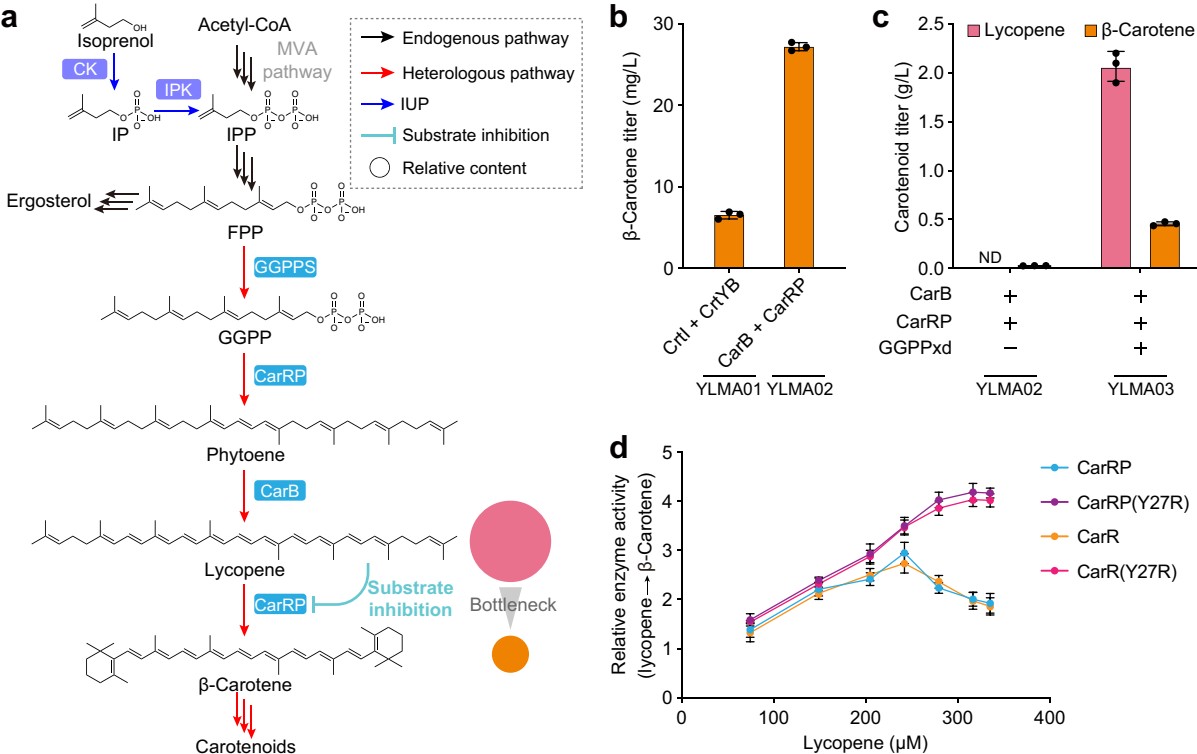

**Fig. 1 lycopene cyclase shows the substrate inhibition effect. a** Lycopene inhibits its downstream enzyme, lycopene cyclase, through substrate inhibition. Consequently, a higher lycopene formation rate than its subsequent conversion rate into β-carotene could aggravate the imbalance, leading to the build-up of lycopene. IP isopentenyl phosphate, IPP isopentenyl diphosphate, FPP farnesyl diphosphate, GGPP geranylgeranyl diphosphate, CK Choline Kinase, IPK Isopentenyl Phosphate Kinase. **b** After 3 days of fermentation in YPD media, β-carotene levels in strains expressing relevant biosynthetic genes from different sources indicated that the CarB/CarRP pair reached higher levels of performance. **c** Heterologous overexpression of GGPP synthase from *X. dendrorhous* (GGPPxd) increased β-carotene production but also led to the major accumulation of lycopene, its biosynthetic precursor. ND not detected. **d** Measurements of relative lycopene cyclase catalytic activity indicated that the activity of wild type CarRP (or CarR) was biphasic with respect to lycopene concentration, indicating substrate inhibition. By contrast, the CarRP (or CarR) variant Y27R was completely free of substrate inhibition. CarR, truncated CarRP without P domain. For **b** to **d**, the average and standard deviation (s.d.) of three biologically independent experiments are shown. Source data are provided as a Source Data file.

showed that lycopene cyclase activity was bi-phasic with respect to lycopene concentration: enzymatic activity increased with lycopene concentration initially, then decreased once lycopene reached higher concentrations (Fig. 1d). This supported the hypothesis that lycopene cyclase was substrate-inhibited, thus creating a major bottleneck in carotenoid biosynthesis.

**Structure-guided protein engineering completely removes substrate inhibition**. Next, we attempted to remove the substrate inhibition effect of lycopene cyclase through protein engineering. As its crystal structure was unavailable, we opted to use the Transform-restrained Rosetta (TrRosetta) platform[21] to create a computational model of the R domain (lycopene cyclase) of CarRP (Supplementary Fig. 6). We employed evolutionary information by generating a Position Specific Scoring Matrix (PSSM) from multiple sequence alignment to identify positions that could be mutated to deconvolute the areas of the enzyme that impacted substrate inhibition. Single and double amino acid substitutions were created based on the PSSM information and clustered to ensure the maximum spread of the tested variants (Supplementary Fig. 7). The variants were clustered using PAM30 to compute distances between sequences and agglomerative clustering to subdivide the sequences, maximizing information obtained during the initial screen. Through this method, we generated a set of 50 candidates with mutations spread throughout the enzyme (Fig. 2a). The selectivity for β-carotene

were obtained for each of the variants (Fig. 2b). Among them, three variants, Y27R, V175W, and T31R-F92W, displayed significantly increased β-carotene selectivity as well as improved production metrics (Fig. 2b, c) without affecting gene expression (Supplementary Fig. 8), suggesting alleviation of the substrate inhibition effect. The substitutions in all three variants were located in a specific part of the enzyme (Supplementary Fig. 9), with Y27R being the most pronounced for loss of inhibition. The variant Y27R demonstrated a complete loss of substrate inhibition without reduction in enzyme activity (Fig. 1d), and yielded a titer of 2.38 g/L of β-carotene (Fig. 2c), along with a selectivity of 98% (compared to 18% of the wild type, Fig. 2d).

We then investigated whether the β-carotene pathway containing variant Y27R was able to maintain its properties of minimal substrate inhibition in the presence of considerably higher precursor/substrate formation rates. To this end, we overexpressed four key enzymes, tHMGR, ERG12, IDI, and ERG20, of the Mevalonate (MVA) pathway[22,23] (Supplementary Fig. 1) in our β-carotene producing strain YLMA11 expressing Y27R (Fig. 2e). This resulted in an increased titer of 3.43 g/L β-carotene while maintaining the high selectivity of 97.8%. In addition, the synthetic Isopentenol Utilization Pathway (IUP)[24–27] was introduced through the expression of Choline Kinase (CK) and Isopentenyl Phosphate Kinase (IPK) (Supplementary Fig. 1), resulting in an additional 23% increase in β-carotene production (4.22 g/L), without any loss in selectivity (YLMA15, Fig. 2e). These results demonstrate that

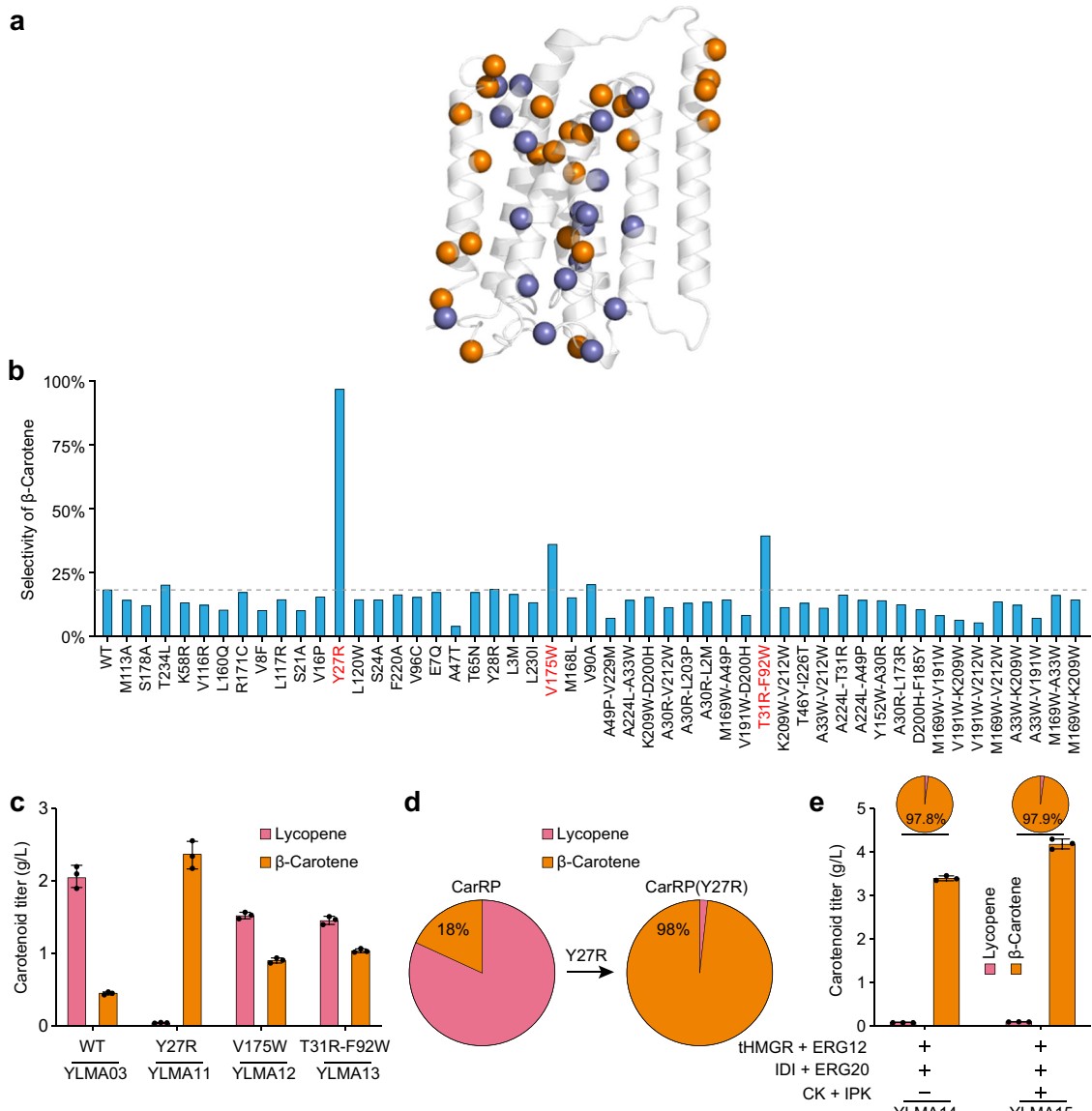

**Fig. 2 Abolishment of substrate inhibition through protein engineering. a** Using a predicted protein model, several positions within the R domain (lycopene cyclase) of CarRP were identified as suitable locations for mutation in order to reduce substrate inhibition. Single substitutions are indicated by copper spheres whereas double substitutions are indicated by purple spheres. **b** β-carotene selectivity was tested on a total of 50 generated variants. Compared to wild type (WT), Y27R, V175W, and T31R-F92W (in red) showed the significantly increased β-carotene selectivity, suggesting a reduction in substrate inhibition. Data represent the mean value of two independent experiments. **c**, **d** Compared to the control strain YLMA03 harboring wild-type CarRP, the variants showed significantly increased production of β-carotene, along with a decrease in lycopene accumulation (**c**). In particular, the strain YLMA11 expressing CarRP (Y27R) achieved a titer of 2.38 g/L (**c**), in addition to a high selectivity of 98% (**d**). **e** The abolishment of substrate inhibition allowed higher fluxes to be channeled through the carotenoid synthesis pathway (through MVA and IUP overexpression), improving β-carotene titers while maintaining the high selectivity. Ultimately, 4.22 g/L β-carotene was produced in YLMA15 with ~98% selectivity. For cultures using strains containing IUP, 30 mM isoprenol (Supplementary Fig. 10) was added to the media post-glucose depletion. For **c** to **e**, the average and s.d. of three biologically independent experiments are shown. Source data are provided as a Source Data file.

the reconstituted non-substrate inhibition pathway functions efficiently in high isoprenoid flux strains, not affected by the intracellular levels of precursor/substrate in the cells.

**Managing substrate inhibition of lycopene cyclase through a GGPPS-mediated flux flow restrictor**. We also explored other viable options that can eliminate the substrate inhibition without the need to modify lycopene cyclase. We hypothesized that attenuating the formation rate of lycopene relative to

its conversion rate could potentially lower the intracellular concentration of lycopene below the inhibitory level. However, this needs to be well-tuned to prevent an overall reduction of the production rate of end-product by attenuating too much the lycopene formation rate. To this end, we exploited the branching point at the FPP node to create a metabolic flow restrictor and regulate flux towards lycopene such as to maintain sub-inhibitory levels, yet high lycopene conversion into β-carotene (Fig. 3a). As such, we looked for GGPPS mutants with variable activity by screening five different enzymes

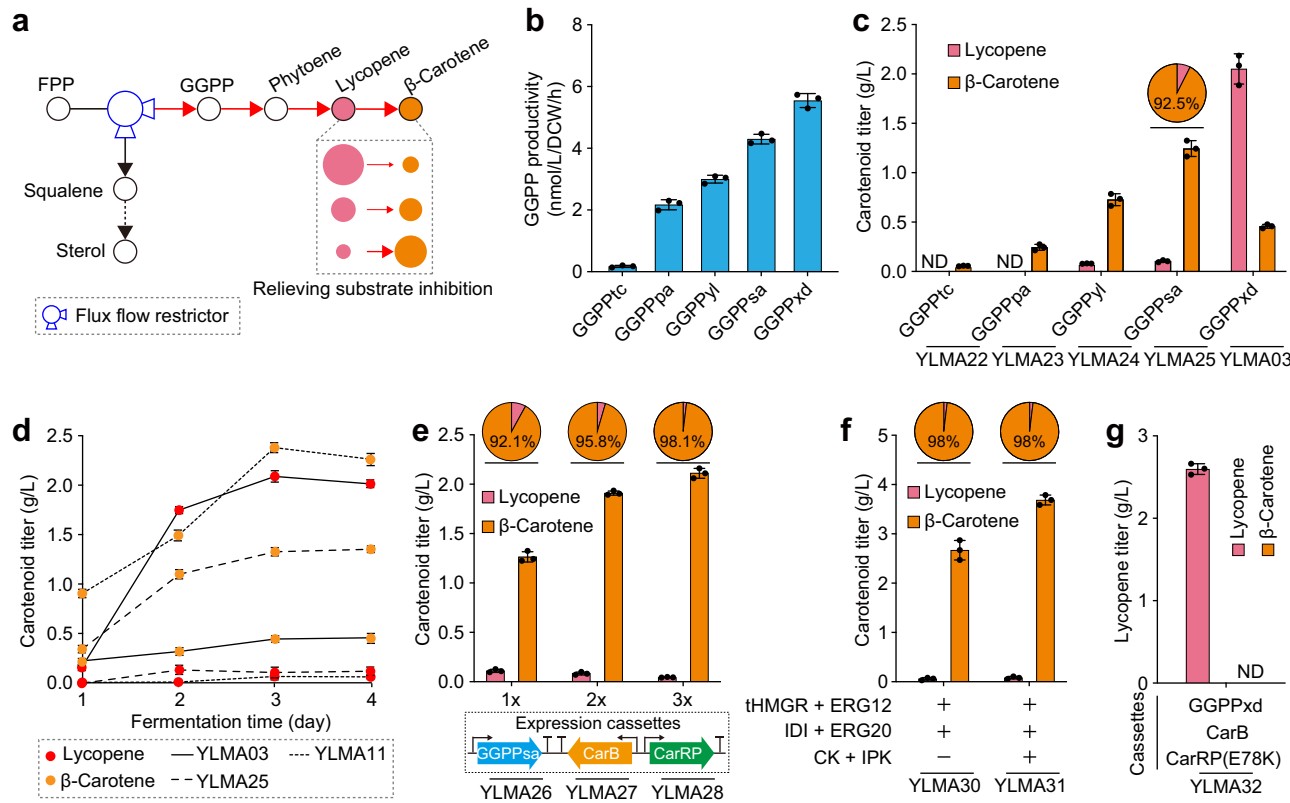

**Fig. 3 A GGPPS-mediated metabolic flow restrictor effectively relieves substrate inhibition. a** A GGPPS-mediated metabolic flow restrictor can vary the amount of flux through the carotenoid synthesis pathway, thus regulating lycopene formation rates. **b** Changes in GGPPS activity can be achieved by expressing enzymes from different organisms in *Y. lipolytica*, as indicated by the varying in vivo GGPP synthesis rate. In these experiments, a po1f background strain with no modifications other than GGPPS expression was used. **c** Compared to the strain expressing GGPPxd, other strains housing lower-activity GGPPSs mitigated the substrate-inhibition effect of lycopene cyclase. The slower formation rates of lycopene prevented its accumulation and hence nearly all lycopene was converted into β-carotene. ND, not detected. **d** Fermentation time courses indicated that a balanced pathway with the attenuated GGPPsa (YLMA25) led to minimal lycopene build-up throughout the experiment, consistent with YLMA11, which contained the Y27R variant of CarRP. On the contrary, rapid lycopene accumulation was observed in the strain with the exceedingly efficient GGPPxd (YLMA03). **e** Gene expression cassettes containing the balanced pathway (GGPPsa, CarB, and CarRP) was sequentially introduced into the po1f-T strain for β-carotene production. With higher copy numbers, β-carotene titers increased up to 2.13 g/L, and the selectivity of β-carotene was maximized. **f** Overexpressing MVA and IUP further improved β-carotene synthesis while maintaining its high selectivity. **g** Using the exceedingly efficient GGPPxd to deliberately trigger substrate inhibition, along with a mutated CarRP(E78K), a lycopene-producing strain was constructed, which reached a titer of 2.62 g/L. ND, not detected. For **b** to **g**, The average and s.d. of three biologically independent experiments are shown. Source data are provided as a Source Data file.

(Supplementary Table 1) with diverging catalytic efficiencies measured by their in vivo GGPP synthesis rates (Fig. 3b). Relative to GGPPxd, the other four GGPPSs exhibited lower productivities (Fig. 3b), which should translate to lower lycopene synthesis flux. Upon introduction of these lower-activity GGPPSs into strain YLMA02 (base strain harboring wild-type CarRP), lycopene levels were reduced (Fig. 3c) and β-carotene production increased, reaching up to 1.26 g/L with 92.5% selectivity when GGPPsa from *Sulfolobus acidocaldarius* was used (Fig. 3c). This was further confirmed by the time courses of lycopene and β-carotene concentrations (Fig. 3d). Interestingly, although the expression of the attenuated GGPPsa (YLMA25) slightly lowered flux committed to carotenoid synthesis, it circumvented any substrate inhibition and enabled an overall balanced pathway that directs all carbon flux towards β-carotene formation, allowing the pathway to behave similar to the variant with Y27R (YLMA11, Fig. 3d). In contrast, strain YLMA03 expressing the very efficient GGPPxd caused the buildup of lycopene at an overly rapid rate, triggering substrate inhibition and preventing its conversion into β-carotene (Fig. 3d). These results indicate that substrate inhibition

could be effectively relieved by the GGPPS-mediated metabolic flow restrictor to regulate flux through both up- and downstream of lycopene for its optimal conversion into β-carotene.

In order to close the gap of β-carotene production between our two engineering strategies that mitigate substrate inhibition (Figs. 2c and 3c), we overexpressed the GGPPS-mediated flux flow restrictor-supported pathway by inserting additional copies (Fig. 3e). This modulation not only strengthened β-carotene biosynthesis up to 2.13 g/L, but also further maximized its selectivity (from 92.1 to 98.1%, Fig. 3e). Furthermore, similar to the previous results, overexpression of MVA and IUP promoted the production of β-carotene up to 3.72 g/L without affecting selectivity (Fig. 3f). This further underlined the fact that the pathway can operate at high efficiency once the substrate inhibition issue has been addressed.

We note that the product profile of β-carotene versus lycopene can be shifted by varying the in vivo GGPPS activity (Fig. 3b, c). We can also take advantage of the high activity of GGPPxd to reconstitute a dedicated lycopene-producing strain. Combining this idea with a CarRP variant (E78K)[20] (Supplementary Fig. 11), we were able to successfully produce 2.62 g/L of lycopene with

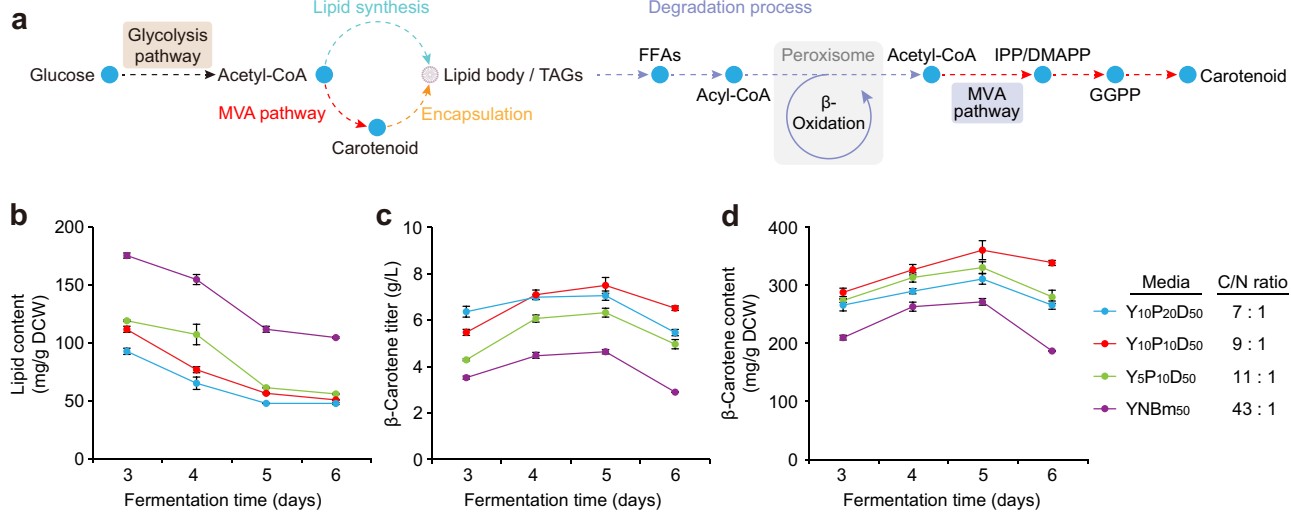

**Fig. 4 Balancing acetyl-CoA distribution between lipid and isoprenoid synthesis benefits carotenoid accumulation. a** Cytosolic acetyl-CoA is shared between two competing pathways, de novo lipid biosynthesis and the MVA pathway. However, lipid bodies within the cell form a hydrophobic region in which carotenoids can be sequestered, thus promoting their accumulation. Consequently, both pathways are necessary, and an optimal partitioning of flux is crucial in achieving high carotenoid titers and per-cell content. In addition, intracellular TAGs can be used as a carbon source for acetyl-CoA generation, which in turn provides the building blocks for carotenoids. TAGs triacylglycerol, FFAs free fatty acids, IPP isopentenyl diphosphate, DMAPP dimethylallyl diphosphate. **b** Lipid content is dependent on the C/N ratio of the media, and a higher C/N ratio promotes lipid production. **c**, **d** β-carotene titer (**c**) and content (**d**) are also functions of the media C/N ratio. However, unlike lipid content, which increased monotonically with the C/N ratio, there is an optimum for β-carotene production. The highest titer and per-cell content of β-carotene occurred at a C/N ratio of 9:1 in $Y_{10}P_{10}D_{50}$ media. For **b** to **d**, The average and s.d. of three biologically independent experiments are shown. Source data are provided as a Source Data file.

undetectable amounts of β-carotene (Fig. 3g). An additional increase in lycopene production was achieved by overexpressing the MVA pathway and introducing IUP, reaching titers of 3.09 g/L (YLMA34, Supplementary Fig. 12).

**Partitioning carbon flux between isoprenoid and lipid synthesis to enhance intracellular carotenoid accumulation.** Lipid bodies in *Y. lipolytica* create hydrophobic pockets that facilitate lipophilic isoprenoid product sequestration and storage[28]. However, while increased triacylglycerol (TAG) supply would enhance isoprenoid storage[29,30], this comes at the expense of acetyl-CoA, a common precursor for isoprenoid and lipid synthesis (Fig. 4a). Therefore, carbon flux needs to be optimally partitioned between lipid and isoprenoid synthesis, with the goal of ensuring a sufficient supply of lipids to encapsulate the produced isoprenoid while not drawing too much acetyl-CoA away from the MVA pathway. To this end, we set out to culture our β-carotene-producing strain YLMA15 in media with varying Carbon-to-Nitrogen (C/N) ratios (Supplementary Table 2), as the capacity for lipid formation can be readily controlled by the C/N ratio of the culture media[31,32]. The initial glucose concentration for all conditions was fixed at 50 g/L, which was determined to be the optimum for our strains (Supplementary Fig. 13). We found that with increasing C/N ratios, the lipid content of the cells increased monotonically (Fig. 4b), while the total biomass decreased (due to reduced nitrogen availability, Supplementary Fig. 14). However, an optimal condition for β-carotene production was obtained in terms of both titer (7.5 g/L, Fig. 4c) and cellular content (360.8 mg/g DCW, Fig. 4d) by using $Y_{10}P_{10}D_{50}$ (10 g/L yeast extract, 10 g/L peptone and 50 g/L glucose) media with a C/N ratio of 9:1. Deviations from this optimum resulted in diminishing β-carotene levels, which was consistent with our hypothesis and highlighted the importance of optimally balancing carotenoid and lipid biosynthesis. The optimal $Y_{10}P_{10}D_{50}$ media was also applied to the lycopene-producing strain (YLMA34), where a

concentration of 8.02 g/L lycopene was obtained after a 5-day fermentation (Supplementary Fig. 15).

**Carotenoid biosynthesis during glucose-depleted stationary phase is supported by cellular lipid degradation.** In our fermentation experiments with strain YLMA15, we found that the content of β-carotene continued to rise during the stationary phase, even after glucose in the media had been depleted (Fig. 5a). It is likely that *Y. lipolytica* mobilized previously stored TAGs as an alternative carbon source to support carotenoid formation[33]. To test this hypothesis, we set out to characterize the cells along with their intracellular lipids both before and after glucose depletion. Throughout the fermentation process, lipid content increased initially, reaching a maximum at day 3, after which it rapidly decreased when glucose was fully consumed (Fig. 5a). Yet, despite the reduction in lipid content, β-carotene content continued to rise well past the point of glucose depletion (Fig. 5a), suggesting that TAGs were utilized to sustain metabolic activity and carotenoid synthesis. Microscopic visualization of the changes that occurred during fermentation was consistent with this hypothesis. When glucose was still present during the initial 3 days, lipid droplets within cells progressively agglomerated into lipid bodies that sequestered the produced β-carotene (Supplementary Fig. 16). However, it was also evident that the lipid bodies were no longer visible during the later stages of fermentation due to TAG breakdown, which in turn caused the accumulated β-carotene to be more dispersed throughout the cell (Supplementary Fig. 16).

Since TAG degradation occurs primarily through β-oxidation to generate acetyl-CoA[34] (Fig. 4a), it is likely that this provides the carbon backbone for β-carotene synthesis during the glucose-depleted stationary phase. To test this hypothesis, we cultured YLMA15 in YNB media with uniformly labeled [U-$^{13}C_6$]glucose and natural abundance stearic acid. Upon the addition of stearic acid, which is catabolized through β-oxidation, we observed large

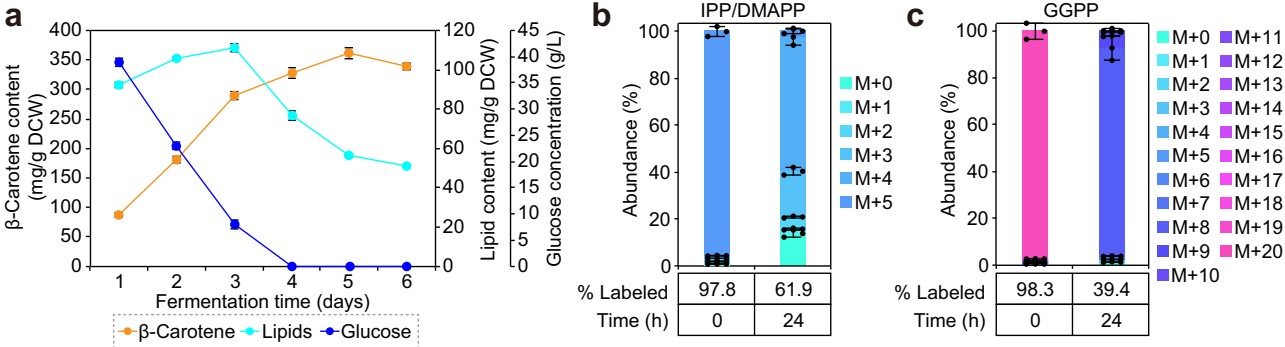

**Fig. 5 Cellular lipids drive carotenoid biosynthesis through β-oxidation during stationary phase after glucose depletion. a** Monitoring glucose concentration, lipid content, and β-carotene content throughout fermentation revealed that β-carotene continued to increase after glucose was exhausted from the media. On the other hand, intracellular lipids rapidly declined post glucose depletion, suggesting that cells were mobilizing TAGs as the alternative carbon source when glucose was no longer available. **b, c** Tracing carbons from cells cultured in [U-$^{13}$C]glucose and natural abundance stearic acid indicated that β-oxidation can be a source for acetyl-CoA destined for carotenoid synthesis. The average and s.d. of three biologically independent experiments are shown. Source data are provided as a Source Data file.

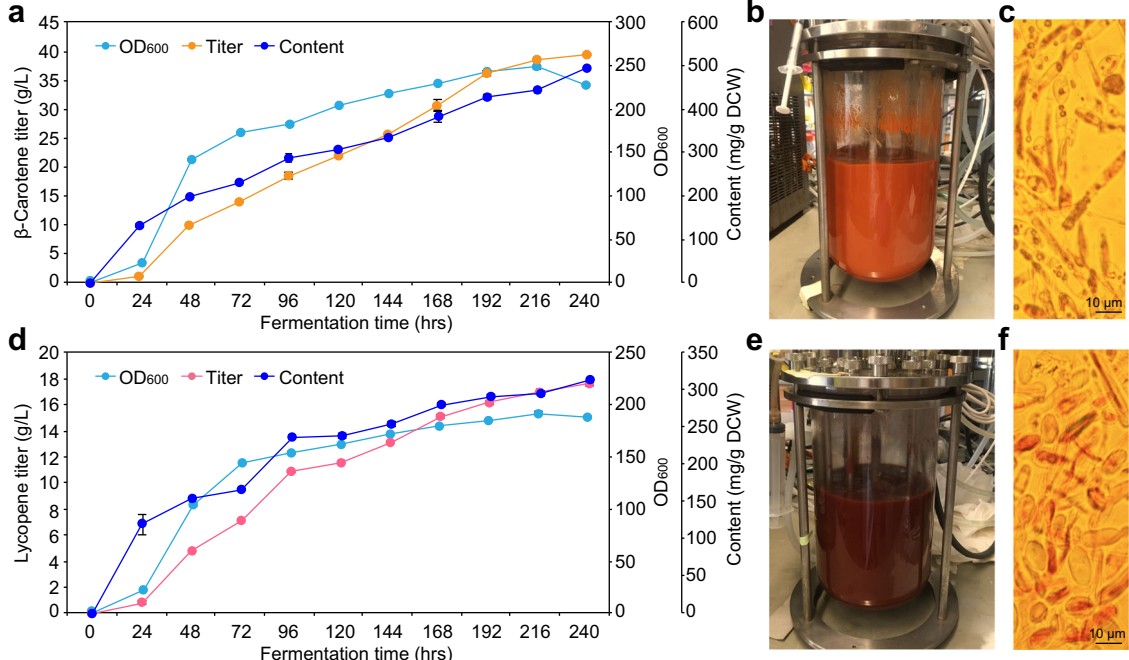

**Fig. 6 Bioreactor fermentation of β-carotene and lycopene engineered Strains. a, d** Fermentation profiles of the β-carotene-producing strain YLMA15 (**a**) and lycopene-producing strain YLMA34 (**d**) in a 3-L bioreactor. **b, e** β-carotene cultures displayed a deep red-orange color after 240 h cultivation (**b**), while lycopene cultures displayed a deep red color (**e**). **c, f** Microscopic images of cells producing β-carotene (**c**) and lycopene (**f**). β-carotene and lycopene are shown to accumulate intracellularly and dispersed throughout the cytoplasm in the most cells. For **a** and **d**, The average and s.d. of three biologically independent experiments are shown. For **c** and **f**, Three independent experiments were repeated with similar results. Source data are provided as a Source Data file.

fractions of unlabeled IPP/DMAPP and GGPP, the main precursors for β-carotene synthesis, within 24 h of cultivation (Fig. 5b, c). These results suggested that the acetyl-CoA formed through β-oxidation could indeed support the MVA pathway and eventually contribute to β-carotene formation. Therefore, this is likely the mechanism used by the cells to convert TAGs into carotenoids during the glucose-depleted stationary phase.

**Bioreactor cultivation studies.** We finally evaluated the performance of the constructed strains alleviated from substrate inhibition in 3-L fed-batch cultivations. After bioreactor optimizations, strain YLMA15 achieved a total β-carotene titer and content of

39.5 g/L and 494 mg/g DCW, respectively, with a productivity of 0.165 g/L/h (Fig. 6a–c). Likewise, bioreactor fermentation of the lycopene-producing strain YLMA34 yielded 17.6 g/L of lycopene (313 mg/g DCW) with a productivity of 0.073 g/L/h (Fig. 6d–f). Notably, we observed no changes in the selectivity for β-carotene in strain YLMA15 during the scale-up process (98% in the fed-batch bioreactor, compared to 97.9% in shake flasks Supplementary Fig. 17). These figures demonstrate the robustness of our engineering strategy in large culture volumes and high-cell density fermentation, with production metrics exceeding previous results (a summary of reported carotenoid production is provided in Supplementary Table 3).

## Discussion

Throughout evolution, host organisms have developed tightly regulated and balanced metabolic systems. Some enzymes are inhibited by their own substrates, which serves to precisely regulate the needs for cell growth. However, when these enzymes are utilized to construct cell factories to overproduce value-added chemicals, substrate inhibition becomes a barrier preventing efficient biotransformation. Unlike most other forms of regulation, the self-reinforcing nature of substrate inhibition amplifies minor differences between metabolite formation and depletion rates, requiring either complete abolishment of the undesired mechanism or careful balancing of the rates.

In this work, we have demonstrated that lycopene cyclase undermines β-carotene production by substrate inhibition, a regulatory effect less reported in the context of microbial synthesis. Substrate inhibition of enzymes could be overcome through modification of the protein structure, a strategy that has been successfully applied to many enzymes[16]. However, these efforts rely on readily available protein crystal structures, which is not the case for the lycopene cyclase investigated here. Although directed evolution is a powerful method of adapting enzymes to specific tasks[35], it often requires high-throughput detection methods to screen large libraries. In addition, due to lack of crystal structure information, efficient structure-driven design heavily relies on both the quality of computational modeling and the accuracy to dock the substrate to its binding site. Here, combining structure and phylogenetic information, sharpened our search and allowed us to isolate a promising mutant by screening only 50 variants, with three of the variants exhibiting diminished or removed substrate inhibition. Furthermore, information from key amino acids could be iteratively fed back into the computational model to further optimize enzyme properties. To understand mechanistically what factor caused the removal of the substrate inhibition would require more thorough investigation that is beyond the scope of this study. The low number of protein variants designed and tested suggests that structure-guided approach coupled with phylogenetic information offers an effective strategy for protein engineering.

The degree of substrate inhibition can also be controlled by tuning the relative rates of up- and downstream pathways forming and consuming the inhibiting substrate. In the case of β-carotene synthesis, selecting GGPPS variants with lower activity reduced the flux through the carotenoid pathway. Yet, the resulting abolishment of substrate inhibition enabled all carotenoid flux to be diverted to β-carotene synthesis, as opposed to a combination of both lycopene and β-carotene. This led to an increased β-carotene production at high specificity (>98%) despite a lower GGPPS activity. On the other hand, substrate inhibition can also be deliberately exploited if lycopene is the desired product. In this case, a highly-efficient GGPPS can cause lycopene formation to outpace its depletion, leading to its accumulation, which then further amplifies the imbalance through substrate inhibition. Correspondingly, the product profile shifts drastically from β-carotene-rich to lycopene-rich. These findings illustrate that engineering proximal enzymes can have profound effects on pathway dynamics, providing a new paradigm for controlling metabolism.

Another important consideration in metabolic engineering is how heterologous pathways interact with the native ones. Designing pathways that are orthogonal to or have minimal impact on the native functions of an organism has been a focal point of many strain engineering efforts[22,36–38]. It is well known that the lipophilic nature of carotenoids promotes their storage in the lipid bodies of the cells. Larroude et al.[30] found that an engineered lipid overproducer strain was capable of producing more β-carotene with a titer of 6.5 g/L, while accompanied with the production of 42.6 g/L lipids. While providing compatible compartments for hydrophobic isoprenoid accumulation, the de novo formation of TAGs also consumes a large amount of carbon source, resulting in limited acetyl-CoA flux into the MVA and product-forming pathway. Here, we demonstrate that *Y. lipolytica*'s native capacity for TAG accumulation is sufficient for carotenoid sequestration. By balancing the flux distribution between carotenoid and lipid synthesis through C/N ratio adjustments, we were able to preserve a larger portion of the acetyl-CoA pool for carotenoid production, achieving higher titers and per-cell content.

The increasing complexity of novel pathways introduced to cells has highlighted the importance of delivering the appropriate amount of flux to exactly satisfy the biosynthetic demands. Hence, while it is tempting to overexpress enzymes with the highest activity with the intention of maximizing pathway flux, precise tuning of gene expression levels and enzyme catalytic rates can sometimes lead to more optimal results, especially when the metabolic network is highly regulated.

## Methods

**Culture conditions and media**. *Escherichia coli* DH5α cells harboring plasmid were cultured in Luria-Bertani (LB) media (BD bioscience) supplemented with corresponding antibiotics (100 μg/mL ampicillin and 50 μg/mL kanamycin) for plasmid propagation at 37 °C for 16 h. All *Yarrowia lipolytica* strains used in this study were cultivated at 30 °C with shaking at 230 rpm. The media used for culturing *Y. lipolytica* strains was prepared as follows. YPD media containing 10 g/L yeast extract (BD bioscience), 20 g/L peptone (BD bioscience), and 20 g/L glucose (Sigma-Aldrich) was used for carotenoid fermentation. YNB media composed of 1.7 g/L yeast nitrogen base without amino acids and ammonium sulfate (YNB, VWR Life Science), 20 g/L glucose, 5 g/L ammonium sulfate (VWR Life Science), 15 g/L agar (BD bioscience), and 0.77 g/L appropriate complete supplement mixture minus uracil, leucine, or tryptophan (Sunrise science products) was used for selecting transformed *Y. lipolytica* strains. The media with varying Carbon-to-Nitrogen (C/N) ratios provided in Supplementary Table 2 were used to regulate the carbon flux distribution between isoprenoid and lipid synthesis for optimal carotenoid fermentation.

**Construction of plasmids and strains**. *E. coli* strain DH5α was selected for plasmid propagation. *Y. lipolytica* po1f strain served as the base strain, and its derivatives and plasmids used in this study are listed in Supplementary Table 4. The primers (synthesized in Sigma-Aldrich) used for plasmid construction are provided in Supplementary Table 5. The restriction enzymes Not1 and Dpn1 were purchased from New England Biolabs (NEB). KAPA HiFi DNA Polymerase with high-fidelity purchased from KapaBiosystems was used for gene amplification for plasmid construction. GoTaq DNA polymerase (Promega) was used for colony PCR identification. PCR fragments were purified using the ZYMO Fragment Recovery Kit (ZYMO research). Plasmids were constructed using Gibson Assembly kit (NEB) followed by transformation into DH5α cells by heat shock. The successfully constructed plasmids were extracted by the QIAprep Spin Miniprep Kit (Qiagen) and then sequenced at Quintara Biosciences. All engineered *Y. lipolytica* strains were constructed by transforming linearized plasmids (Not1 digestion) using the lithium-acetate method. Recombinants were verified by PCR amplification from genomic DNA. The carotenoid biosynthetic genes evaluated in this study were codon-optimized towards *Y. lipolytica* and synthesized by GeneArt (Thermo Fisher Scientific).

**TRP1 disruption in po1f strain using CRISPR-Cas9**. For TRP1 disruption, the CRISPR-Cas9 plasmid[39] containing gRNA (ACGCCGAGGAGTGGTACCGG) targeting the TRP1 (YALI0B07667g) gene of *Y. lipolytica* was transformed into strain po1f using Ura3 as the auxotrophic marker. The strain with tryptophan auxotrophy was obtained by selecting on YNB-Ura and YNB-Ura-Trp plates. After that, the positive clones were inoculated onto YPD plates and sub-cultured three times to lose the CRISPR-Cas9 plasmid, resulting in the po1f-T strain (ura3⁻, leu2⁻, trp1⁻).

**Auxotrophic markers curation by the Cre-loxP system**. In order to rescue *URA3*, *LEU2* and *TRP1* auxotrophic markers, plasmid pYLMA-Cre harboring Cre recombinase was transformed into target *Y. lipolytica* strains. Transformants were cultured on YPD agar plate supplemented with a final concentration of 250 mg/L hygromycin B (Sigma-Aldrich) for antibiotic selection. After 2~3 days of cultivation at 30 °C, colonies were randomly transferred onto a new YPD agar plate containing hygromycin B, and cultured for 1 more day to allow for more successful marker deletions. Markers curation was then confirmed by transferring the colonies onto YNB-Ura, YNB-Leu, and YNB-Trp agar plates, respectively. Successful deletion of all three markers in strains will show a phenotype with uracil, leucine

and tryptophan deficiency. Plasmid pYLMA-Cre in cells was then removed by incubating engineered strains on YPD agar plates at 30 °C for 24 h, with 2~3 repeats.

**Shake flask fermentations**. Single colonies of recombinant strains were picked from plate, inoculated into 2 mL YPD media, and cultivated overnight (16~18 h) at 30 °C and 230 rpm. The culture was then transferred to a 50 mL shake flask containing 10 mL YPD media (initial $OD_{600} = 0.1$), and cultivated at 30 °C with shaking at 230 rpm for 3~5 days. When applicable, 30 mM isoprenol (Sigma-Aldrich) was added into YPD media when glucose of culture was nearly consumed.

**Bioreactor fermentations**. Fed-batch fermentations were performed in a 3 L bioreactor (New Brunswick Bioflo115 system). The initial fermentation was completed with 1 L medium containing 100 g/L glucose, 100 g/L peptone, and 50 g/L yeast extract. The temperature was maintained at 30 °C. The dissolved oxygen was controlled at 20% with an agitation cascade of 250~800 rpm. Air was sparged into fermenter at 2 vvm. The pH was maintained at 6.8 by feeding 5 M NaOH or 5 M HCL. Foam was prevented by the addition of antifoam 204 (Sigma-Aldrich). The fed-batch process was initiated after 48 h of cultivation with the $10 \times Y_{10}P_{10}D_{50}$ media consisting of 100 g/L yeast extract, 100 g/L peptone and 500 g/L glucose. Once the media feeding starting, the agitation and aeration was changed and held constantly at 600 rpm and 0.3 vvm, respectively. Samples were taken every 24 h to measure $OD_{600}$, glucose concentration, and carotenoid titer.

**Quantification of residual glucose in media**. Glucose concentration in media was determined by High-Performance Liquid Chromatography (HPLC, Agilent technologies 1260) equipped with a refractive index detector and a HPX-87H column (Bio-Rad). The 500 μL sample was extracted and centrifuged at 12,000 g for 5 min, and then the supernatant was filtered through 0.2 μm syringe filters prior to injection. The mobile phase consisted of 14 mM sulfuric acid (Sigma-Aldrich) with a flow rate of 0.7 mL/min at 50 °C. The injection volume was 10 μL.

**Lipid extraction and quantification**. The fatty acids synthesized by *Y. lipolytica* including palmitate (C16:0), palmitoleate (C16:1), stearate (C18:0), oleate (C18:1) and linoleate (C18:2) were quantified using a Gas Chromatography coupled to a Flame Ionization Detector (GC-FID). 0.1~1 mL cell culture was extracted from each bioreactor such that the sample contained approximately 1 mg biomass. A centrifugation step at 16,000 g for 10 min was performed and the supernatant discarded. 0.5 mL of a 0.5 M sodium hydroxide-methanol solution (20 g/L sodium hydroxide in anhydrous methanol) was mixed with the cell pellets, followed by the addition of 100 μL internal standards containing 2 mg/mL methyl tridecanoate (Sigma-Aldrich) and 2 mg/mL glyceryl triheptadecanoate (Sigma-Aldrich) dissolved in hexane. Methyl tridecanoate was used for volume loss correction during sample preparation and glyceryl triheptadecanoate was used for transesterification efficiency correction. The samples were vortexed for 1 h to allow for the transesterification of lipids to fatty acid methyl esters (FAMEs). Afterwards, 40 μL of 98% sulphuric acid (Sigma-Aldrich) was added to neutralize the pH. The FAMEs were then extracted through the addition of 0.5 mL hexane followed by vortexing for 30 min. Centrifugation at 12,000 g for 1 min was then performed to remove cellular debris and the top hexane layer was extracted for analysis. Separation of the FAME species was achieved on an Agilent HP-INNOWax capillary column. The injection volume was 1 μL, split ratio was 10, and the injection temperature was 260 °C. The column was held at a constant temperature of 200 °C and helium was used as the carrier gas with a flow rate of 1.5 mL/min. The FID was set at a temperature of 260 °C with the flow rates of helium make up gas, hydrogen, and air at 25 mL/min, 30 mL/min, and 300 mL/min, respectively.

**Intracellular metabolites extraction and quantification**. To extract intracellular metabolites (e.g., IPP/DMAPP, and GGPP), 1 mL culture was filtered through a 25-mm 0.2 μm nylon filter using vacuum filtration. The cells were washed immediately with 2 mL of water preheated to 30 °C, and the filter was submerged in ice-cold extraction buffer (40% methanol + 40% acetonitrile + 20% water). After incubation at −20 °C for 20 min, the extract solution was centrifuged at 16,000 g for 10 min, and the supernatant was transferred to a new tube and dried. The sample was resuspended with 50 μL water, and then centrifuged at 16,000 g for 10 min. Metabolites in the supernatant were quantified by liquid chromatography-tandem mass spectrometry (LC-MS/MS) comprised of an Agilent 1100 series LC system and an AB Sciex API-4000 MS. 10 μL sample was injected and separation was achieved on a Waters XBridge C-18 column with a mobile phase consisting of solution A (0.1% tributylamine, 0.12% acetic acid, 0.5% 5 M NH₄OH in water, v/v) and solution B (100% acetonitrile). The flow rate was 0.3 mL/min and the following gradients were used: 0–5 min, 0% B; 5–20 min, 0~65% B; 20–25 min, 65% B; 25–30 min, 100% B; 30–35 min, 100% B; 35–36 min 100~0% B, 0% B until 45 min. The analytes were then analyzed using Analyst 1.6.2 and MAVEN 707, and compared to standard curves generated using chemical standards purchased from Sigma-Aldrich and Cayman Chemicals.

**Labeling experiments**. Strains used in labeling studies were revived in YNB media with [U-¹³C]glucose as the sole carbon source. They were then subcultured in the same media and grown until early stationary phase at 30 °C. Samples were taken before the start of the pulse addition of an extra carbon source using the same intercellular metabolite extraction method. Afterwards, 10 mM stearic acid was added to the corresponding cultures, and measurements of metabolite isotopic enrichments were taken at different time points. The optical densities associated with each sample were also recorded. IPP/DMAPP, and GGPP were quantified by LC-MS/MS. All MS data from labeling experiments were corrected for natural abundance using IsoCor[40].

**Extraction of carotenoids**. Carotenoid extraction was performed as described[41] with the following modification. Briefly, 100 μL culture was centrifuged for 1 min at 16,000 g, and cell pellets were suspended in 900 μL dimethyl sulfoxide (DMSO, Sigma-Aldrich) prior to heating at 50 °C for 1 h until the cells bleached in a water bath. The DMSO extracts were briefly mixed with 450 μL of methanol and centrifuged at 16,000 g for 5 min. The resultant supernatants were transferred into 96-well assay plates or glass vials for carotenoid analysis and quantification.

**Analysis and quantification of carotenoids**. The production of carotenoids was expressed as grams per liter of fermentation broth (g/L) and milligrams per gram of dry cell weight (mg/g DCW). Optical densities were measured at 600 nm with Thermo Spectronic Genesys 20 (Thermo Scientific) and used to calculate cell mass ($DCW = 0.35 \times OD_{600}$ for β-carotene and $DCW = 0.30 \times OD_{600}$ for lycopene, Supplementary Fig. 18). The analysis and quantification of β-carotene was performed by HPLC (SHIMADZU LC-20 AT) equipped with a Kromasil C18 column (4.6 mm × 250 mm) and UV/VIS detection at 450 nm. The mobile phase consisted of acetonitrile-methanol-isopropanol (5:3:2 v/v) with a flow rate of 1 mL/min at 40 °C. The analysis and quantification of lycopene were performed with Spectramax M2e Microplate Reader (Molecular devices) or HPLC at 470 nm. The standard curves of β-carotene and lycopene (Sigma-Aldrich) were prepared by running the same extraction process as the samples.

**Quantitative real-time PCR**. Real-time PCR (RT-PCR) was used to estimate the relative gene expression. mRNA extracted by MasterPure™ Yeast RNA purification kit (Lucigen, Wisconsin, USA) was used as the template. RT-PCR was carried out on an iCycler (Bio-Rad, USA) using iScript™ one-step RT-PCR kit with SYBR Green Supermix (Bio-Rad, USA) according to the manufacturer's instructions. ACT1 was used as an internal control gene for normalization. The relative gene expression was calculated using the comparative $2^{-\triangle\triangle CT}$ or $2^{-\triangle CT}$ method.

**In vitro enzymatic assays**. Yeast microsomes for in vitro enzymatic assays were prepared as described previously[42]. Briefly, strains harboring wild type or mutated CarRP were grown overnight in YNB media at 30 °C and then inoculated into 200 mL YNB media to an initial $OD_{600}$ of 0.1. After 24 h cultivation, cells were collected by centrifugation at 4,000 g for 10 min. Resuspension of the cells in TEK buffer (50 mM Tris-HCl, pH 6.8, 1 mM EDTA, 0.1 M KCl) followed, and the solution was kept at room temperature for 5 min. Afterwards, the cells were recovered by centrifugation, washed in TES buffer (50 mM Tris-HCl, pH 6.8, 1 mM EDTA, 0.6 M sorbitol), resuspended in TESM buffer (50 mM Tris-HCl, pH 6.8, 1 mM EDTA, 0.6 M sorbitol, 14 mM 2-mercaptoethanol), and left at room temperature for 10 min. Then, the cells were recovered once again by centrifugation, washed in extraction buffer (50 mM Tris-HCl, pH 6.8, 1 mM EDTA, 1 mM PMSF), and resuspended in extraction buffer. Glass beads were added to each sample, which were intermittently vortexed for 30 s and placed on ice for 30 s for a total of 15 repeats. The cell pellets were then discarded by centrifugation at 4,000 g, 4 °C for 10 min, and the supernatant was transferred to a 50 mL tube. The crude yeast microsomal fraction collected above was used for in vitro assays of lycopene cyclase. Standard enzyme assays were performed in a total volume of 200 μL containing 50 mM Tris-HCl (pH 6.8), 1 mM phenylmethylsulfonyl fluoride (PMSF) and 1 mg of microsomal protein. Serial concentrations of lycopene (50~350 μmol/L) dissolved in dimethyl sulfoxide (DMSO) were used as the substrate. Reactions were initiated by substrate addition, incubated at 30 °C with gentle shaking for 16 h, and then terminated by adding 200 μL ethyl acetate. The solution was vortex for 10 min, and the organic phase was collected by centrifugation and analyzed by HPLC.

**Generation of protein variants**. The variants were generated by analyzing the amino acid conserved and co-evolutionary information of this protein family from Position Specific Scoring Matrix (PSSM). Here we generate the matrix using psi-blast from ncbi-blast-2.7.1 + [43] with uniref90[44] as the database (https://www.uniprot.org/help/uniref) and an *E*-value of 0.01 running 3 iterations. For all positions in the protein, the PSSM score which represents conservatism of amino acids was calculated for both lycopene cyclase and other homologous proteins of this family. The higher score indicates the more conservative of the amino acid in this position. We screened the substitutions that could be replaced with more conserved amino acids based on the PSSM scores. The different value between the potential substitutions and wild-type amino acids was calculated and the scores were sorted. All glycine substitutions were removed from the scoring. Top

25 scoring substitutions were combined into double substitutions randomly. A distance matrix was computed using the PAM30 substitution matrix and clustered using Agglomerative in sklearn[45] into 25 clusters to minimize the number for test. The variants were chosen randomly within each cluster. The ranked 26–50 scoring substitutions were ordered as single mutational variants.

**Homology model of lycopene cyclase**. A homology model was generated of the lycopene cyclase using TrRosetta server[21] submitted the sequence of the R domain (1–239 amino acids) of CarRP.

**Calculation of the C/N ratio in media**. The C/N ratio was calculated by referring to the composition of Yeast extract (Bacto™) and Peptone (Bacto™) in the BD Bionutrients™ technical manual (Supplementary Table 6) (https://legacy.bd.com/ds/technicalCenter/misc/lcn01558-bionutrients-manual.pdf). The carbon in Yeast extract and Peptone was ignored because of its extremely lower concentration relative to that of glucose. The total nitrogen in Yeast extract and Peptone is 10.9% and 15.4%, respectively. The C/N ratio was generated with the following formula **1**. X, Y, and Z represent the concentration of glucose, yeast extract and peptone respectively.

$$\frac{Carbon\ of\ glucose}{Nitrogen\ of\ (yeast\ extract + peptone)} = \frac{\frac{X}{180.156\,g\,mol^{-1}} \times 6}{\frac{(10.9\% \times Y + 15.4\% \times Z)}{14\,g\,mol^{-1}}} \tag{1}$$

**Reporting summary**. Further information on research design is available in the Nature Research Reporting Summary linked to this article.

## Data availability
All data supporting the findings of this study are available within the paper and its supplementary information files. A reporting summary for this article is available as a supplementary information file. The uniref90 database used in this study are available at https://www.uniprot.org/help/uniref. Source data are provided with this paper.

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

## Acknowledgements
This work was funded by the National Key Research and Development Program of China (2018YFA0901800), and the DiSTAP Center of the Singapore-MIT Alliance for Research and Technology.

## Author contributions
Y.M., S.H., and G.S. conceived and designed research; Y.M. performed research; P.G. designed protein variants; Y.M., N.L., and J.L. analyzed data; Y.M., N.L., P.G., J.L., K.Q., and G.S. wrote the paper.

## Competing interests
The authors declare no competing interests.
