## [Peer Review File · Nature Communications]

Removal of lycopene substrate inhibition enables high carotenoid productivity in *Yarrowia lipolytica*Reviewers' Comments:

Reviewer #1:

Remarks to the Author:

Report on "Removal of substrate inhibition enables high carotenoid productivity in *Yarrowia lipolytica*" by Ma Y. et al.

The paper by Ma Y. et al. described the optimized production of β -carotene (and lycopene) by the yeast *Yarrowia lipolytica* using two different strategies in order to ensure high yield and productivity. One was to test structure-based guided mutations of lycopene cyclase of *Mucor circinelloides* to relieve substrate inhibition, the other was based on a flux balance analysis to stay under the substrate concentration leading to inhibition of *M. c.* lycopene cyclase. Both strategies provide equivalent yield as well as equivalent productivities in β -carotene. An artificial terpene biosynthetic pathway was also introduced to enhance final yields in carotenoids. Thereafter the bioreactor-based carotenoid production was studied, reaching the highest reported yield ever described.

The job as a whole is well done, the planned objectives ambitious, the bottlenecks well identified and elegantly removed leading to particularly attractive carotenoid yields. The two strategies used to counter the deleterious effects of substrate inhibition detected for lycopene cyclase from *Mucor circinelloides* could be used for other enzymes and will therefore be of great interest to the community. The data provided are sufficient and in line with the conclusions of the different parts of the manuscript. The article is well organized and easy to follow, with the exception of a few figures and despite a few other minor points to correct deserves publication in *Nature Communications*.

Minor remarks to be addressed:

- Showing the chemical formulas of GGPP, phytoene, lycopene, beta-carotene, isopentenol and dimethylallyl alcohol would be beneficial to reach a wider audience
- Fig. 4b, 4c and 4d as well as supplementary Fig. 15 are difficult to catch immediately. It is suggested to change the x axis putting the days instead of the C/N ratio and choosing a color for each C/N ratio and thus expressing the OD600 in function of time for the different C/N ratios.
- There is a lack of experimental details in the section *In vitro* enzymatic assays. Buffer, pH, form of added substrate (dissolved? which solvent? which concentration of substrate and solvent?) and other relevant details should be provided.
- The calculation of the C/N ratio should be more detailed (to be put in the supplementary part). In particular, the C/N ratio for the YNBm50 medium of 43/1 seems to be much too high, but difficult to be affirmative without an explanation of the calculation.
- Line 621: change wide for wild.
- Fig. 4a: TCA cycle should be replaced by glycolysis (+ pyruvate dehydrogenase).

Reviewer #2:

Remarks to the Author:

The authors presented successful results of b-carotene production by removal of substrate inhibition of lycopene cyclase of CarRP from *M. circinelloides*.

The substrate inhibition of lycopene cyclase is a novel finding. Thus, it needs to be supported by strong evidence. The reviewer wonders whether it is restricted to only CarRP. It is required to do the *in vitro* experiment with other lycopene cyclases than the CarRP. The authors also need to show the mutation effect of CarRP for removal of the substrate inhibition is applied to the truncated CarR too. The reviewer also wonders whether the yeast microsomal system is appropriate for the *in vitro* assays of lycopene cyclase. Are there any previous reports using the methods for the analysis? If not, the authors should describe the method in detail for the readers. As the authors know, the lycopene substrate and b-carotene product are highly hydrophobic compounds insoluble in the aqueous phase. The accumulation of lycopene has been rarely reported in the metabolic engineering of other hosts. The reviewer wonders whether lycopene cyclases are overexpressed sufficiently compared to other carotenoids enzymes such as lycopene synthase. The authors need to measure the amounts of the proteins involved in the b-carotene synthesis.

Reviewer #3:

Remarks to the Author:

Comments:

- The manuscript identifies substrate inhibition of lycopene cyclase as the major bottleneck in β -carotene biosynthesis pathway in *Y. lipolytica*. This bottleneck was eliminated almost completely using experimental and computational methods. The GGPPS mediated flux valve and the protein engineering approach prove successful in removing substrate inhibition. These strategies could be possibly extrapolated for the synthesis of other natural products in *Y. lipolytica*.
- The manuscript is well structured with a clear explanation of hypothesis and experimental evidence supporting the hypothesis.
- The findings are important and unique in their contribution to the overall metabolic engineering field. The ID and removal of substrate inhibition are likely to be as widespread as the use of the truncated HMGR.
- Could the authors please expand on the discussion of the protein engineering design method. From the manuscript text alone, it is difficult to understand exactly how the PSSM was used for identifying potential mutation sites.
- What were the results of the in vitro activity of the 47 mutants not described in Figure 2b/c?
- The term flux valve seems a bit misleading. Maybe this is more appropriately a flux flow restrictor? The work described seems to be flux tuning. A valve can be opened and closed and adjusted but there is only one state for the "valve" in each design.
- Lines 98 and 99: The manuscript talks about introducing an additional copy of GGPPS from *X dendrorhous*. Why was *X dendrorhous* GGPPS chosen over *M. circinelloides* GGPPS even though the strain harboring *M. circinelloides* β -carotene synthetic genes produced 4.12-fold more β -carotene? *M. circinelloides* GGPPS may have expressed better and resulted in even higher titers.
- Lines 168 and 169: The manuscript mentions the use of lower activity GGPPS for achieving high β -carotene titers. Decreasing GGPPS activity may result in the accumulation of FPP. FPP accumulation may result in a competing pathway to produce ergosterol. Was any kind of analytical quantification performed to identify FPP/ergosterol?
- It would be better to add the IUP pathway shown in SF1 in the main paper figures.
- Is the improvement of titer upon addition of the IUP pathway (SF12) statistically significant? It does not appear so by eye.
- Please provide the data to support how 30 mM isoprenol and 10 mM prenil was found to be optimal.

Response to Reviewers' comments

Reviewer #1

Report on “Removal of substrate inhibition enables high carotenoid productivity in *Yarrowia lipolytica*” by Ma Y. et al. The paper by Ma Y. et al. described the optimized production of β -carotene (and lycopene) by the yeast *Yarrowia lipolytica* using two different strategies in order to ensure high yield and productivity. One was to test structure-based guided mutations of lycopene cyclase of *Mucor circinelloides* to relieve substrate inhibition, the other was based on a flux balance analysis to stay under the substrate concentration leading to inhibition of M. c. lycopene cyclase. Both strategies provide equivalent yield as well as equivalent productivities in β -carotene. An artificial terpene biosynthetic pathway was also introduced to enhance final yields in carotenoids. Thereafter the bioreactor-based carotenoid production was studied, reaching the highest reported yield ever described.

The job as a whole is well done, the planned objectives ambitious, the bottlenecks well identified and elegantly removed leading to particularly attractive carotenoid yields. The two strategies used to counter the deleterious effects of substrate inhibition detected for lycopene cyclase from *Mucor circinelloides* could be used for other enzymes and will therefore be of great interest to the community. The data provided are sufficient and in line with the conclusions of the different parts of the manuscript. The article is well organized and easy to follow, with the exception of a few figures and despite a few other minor points to correct deserves publication in Nature Communications.

Minor remarks to be addressed:

- Showing the chemical formulas of GGPP, phytoene, lycopene, beta-carotene, isopentenol and dimethylallyl alcohol would be beneficial to reach a wider audience

Response:

Chemical formulas have been added to the revised manuscript.

- Fig. 4b, 4c and 4d as well as supplementary Fig. 15 are difficult to catch immediately. It is suggested to change the x axis putting the days instead of the C/N ratio and choosing a color for each C/N ratio and thus expressing the OD600 in function of time for the different C/N ratios.

Response:

We thank Reviewer 1 for this suggestion and have revised the Figures accordingly as follows.

Fig. 4b, 4c and 4d

Supplementary Fig. 15

- There is a lack of experimental details in the section In vitro enzymatic assays. Buffer, pH, form of added substrate (dissolved? which solvent? which concentration of substrate and solvent?) and other relevant details should be provided.

Response:

Additional experimental details have been provided in the revised manuscript as indicated below in italics.

“.....Standard enzyme assays were performed in a total volume of 200 μ L containing 50 mM Tris-HCl (pH 6.8), 1 mM phenylmethylsulfonyl fluoride (PMSF) and 1 mg of microsomal proteins. Serial concentrations of lycopene (50~350 μ mol/L) dissolved in dimethyl sulfoxide (DMSO) were used as the substrate. Reactions were initiated by substrate addition, incubated at 30 °C with gentle shaking for 16 h, and then terminated by adding 200 μ L ethyl acetate.....”

- The calculation of the C/N ratio should be more detailed (to be put in the supplementary part). In particular, the C/N ratio for the YNBm50 medium of 43/1 seems to be much too high, but difficult to be affirmative without an explanation of the calculation.

Response:

The C/N ratio was calculated by referring to the composition of Yeast extract (Bacto™) and Peptone (Bacto™) in the BD Bionutrients™ technical manual. The details of media composition have been provided in the Supplementary Table 5, and an explanation of the calculation was also provided in the Methods section of the revised manuscript as indicated below in italics.

“Calculation of the C/N ratio in media. The C/N ratio was calculated by referring to the composition of Yeast extract (Bacto™) and Peptone (Bacto™) in the BD Bionutrients™ technical manual (Supplementary Table 5) (<https://legacy.bd.com/ds/technicalCenter/misc/lcn01558-bionutrients-manual.pdf>). The carbon in Yeast extract and Peptone was ignored because of its extremely lower concentration relative to that of glucose. The total nitrogen in Yeast extract and Peptone is 10.9% and 15.4%, respectively. The C/N ratio was generated with the following formula. X, Y, and Z represent the concentration of glucose, yeast extract and peptone respectively.”

$$\frac{\text{Carbon of glucose}}{\text{Nitrogen of (yeast extract + peptone)}} = \frac{\frac{X}{180.156 \text{ g mol}^{-1}} \times 6}{\frac{(10.9\% \times Y + 15.4\% \times Z)}{14 \text{ g mol}^{-1}}}$$

- Line 621: change wide for wild.

Response:

We apologize for this typo and have corrected the manuscript.

- Fig. 4a: TCA cycle should be replaced by glycolysis (+ pyruvate dehydrogenase).

Response:

We have revised the Figures accordingly.

Reviewer #2

The authors presented successful results of b-carotene production by removal of substrate inhibition of lycopene cyclase of CarRP from *M. circinelloides*. The substrate inhibition of lycopene cyclase is a novel finding. Thus, it needs to be supported by strong evidence. The reviewer wonders whether it is restricted to only CarRP. It is required to do the *in vitro* experiment with other lycopene cyclases than the CarRP.

Response:

Both our *in vitro* (Fig. 1d) and *in vivo* (Fig. 3b and c) results demonstrate that substrate inhibition exists within CarRP. As shown in Fig. 3b and c, substrate inhibition emerges after *in vivo* GGPPS activity exceeds a threshold, drastically changing the product profile made by the cells. Additionally, engineering strategies specifically designed to target this substrate inhibition were successful in shifting the product profile towards β -carotene. This evidence, we believe, is sufficient to support our finding of substrate inhibition within CarRP. The concern expressed as to whether the substrate inhibition discovered in lycopene cyclase of CarRP is common in this family is an interesting topic, but beyond the scope of this study. The fungi enzymes CarB and CarRP used in this study are the most popular in yeast engineering (Larroude et al., 2018; Celinska et al., 2017; Gao et al., 2017; Gao et al., 2014; Czajka et al., 2018; Zhang et al., 2020) for carotenoid synthesis because of their better performance relative to bacterial-sourced enzymes. Since CarRP exhibited better performance than other lycopene cyclases in the current manuscript, our experiments are primarily focused on CarRP.

The authors also need to show the mutation effect of CarRP for removal of the substrate inhibition is applied to the truncated CarR too.

Response:

We performed the *in vitro* assay with truncated CarR-mutant (Truncated CarR), and the whole CarRP-mutant [CarRP(Y27R)] and wild type CarRP as controls. The results below showed that the truncated CarR-mutant also can relieve substrate inhibition, similar to the whole CarRP-mutant.

The reviewer also wonders whether the yeast microsomal system is appropriate for the *in vitro* assays of lycopene cyclase. Are there any previous reports using the methods for the analysis? If not, the authors should describe the method in detail for the readers. As the authors know, the lycopene substrate and b-carotene product are highly hydrophobic compounds insoluble in the aqueous phase.

Response:

Yeast microsomal system is often used to study the function of proteins with transmembrane helices, such as cytochromes P450s (Denis et al., 1996; Guo et al., 2013; Li et al., 2019;) and transporters (Yamanka et al., 2005; Zhao et al., 2009; Zhao et al., 2011). Since CarRP is a transmembrane protein with six transmembrane helices (Supplementary Fig. 5), and thus very difficult to isolate and purify the protein, we chose the yeast microsomal system as an appropriate tool for *in vitro* assays. Additional details on this assay has been provided in the revised manuscript.

The accumulation of lycopene has been rarely reported in the metabolic engineering of other hosts. The reviewer wonders whether lycopene cyclases are overexpressed sufficiently compared to other carotenoids enzymes such as lycopene synthase. The authors need to measure the amounts of the proteins involved in the β -carotene synthesis.

Response:

It is possible that prior efforts in β -carotene synthesis did not trigger lycopene accumulation because lower-activity GGPPS were used. It is also possible that the concentration of lycopene was not measured or reported in prior work as it is a side product. Hence, the issue of substrate inhibition by lycopene was largely overlooked. In fact, a previous article focusing on β -carotene production in yeast reported the lycopene accumulation (28.32% of lycopene, 32.56% of β -carotene, and 39.06% of other carotenes) (Xie et al., 2014). Additionally, in order to rule out the possibility of insufficient expression of enzymes, we attempted to enhance the expression of lycopene cyclase by increasing its gene copy number while also using the strong TEF promoter (Supplementary Fig. 3). However, the issue of lycopene accumulation remained unaddressed. By contrast, introducing a lower-activity lycopene synthase with a lower lycopene synthase rate led to improved β -carotene production (Fig. 3b and c). These results suggest that protein expression of lycopene cyclase was not the bottleneck in our study. Our findings showed that substrate inhibition was the main limitation resulting in lycopene accumulation. It is not necessary to measure the amounts of the proteins.

Reviewer #3

Comments:

The manuscript identifies substrate inhibition of lycopene cyclase as the major bottleneck in β -carotene biosynthesis pathway in *Y. lipolytica*. This bottleneck was eliminated almost completely using experimental and computational methods. The GGPPS mediated flux valve and the protein engineering approach prove successful in removing substrate inhibition. These strategies could be

possibly extrapolated for the synthesis of other natural products in *Y. lipolytica*. The manuscript is well structured with a clear explanation of hypothesis and experimental evidence supporting the hypothesis. The findings are important and unique in their contribution to the overall metabolic engineering field. The ID and removal of substrate inhibition are likely to be as widespread as the use of the truncated HMGR.

- Could the authors please expand on the discussion of the protein engineering design method. From the manuscript text alone, it is difficult to understand exactly how the PSSM was used for identifying potential mutation sites.

Response:

Additional detail has been provided in the Methods section of the revised manuscript as indicated below in italics.

*“**Generation of protein variants.** The variants were generated by analyzing the amino acid conserved and co-evolutionary information of this protein family from Position Specific Scoring Matrix (PSSM). Here we generate the matrix using psiblast from ncbi-blast-2.7.1+ with uniref90 as the database and an E-value of 0.01 running 3 iterations. For all positions in the protein, the PSSM score which represents conservatism of amino acids was calculated for both lycopene cyclase and other homologous proteins of this family. The higher score indicates the more conservative of the amino acid in this position. We screened the substitutions that could be replaced with more conserved amino acids based on the PSSM scores. The different value between the potential substitutions and wild-type amino acids was calculated and the scores were sorted. All glycine substitutions were removed from the scoring. Top 25 scoring substitutions were combined into double substitutions randomly. A distance matrix was computed using the PAM30 substitution matrix and clustered using Agglomerative in sklearn into 25 clusters to minimize the number for test. The variants were chosen randomly within each cluster. The ranked 26-50 scoring substitutions were ordered as single mutational variants.”*

- What were the results of the in vitro activity of the 47 mutants not described in Figure 2b/c?

Response:

The *in vitro* activity of these mutants was not investigated. Since the Y27R mutant was experimentally shown to nearly abolish the substrate inhibition effect (Fig. 2b and c), we did not think it was necessary to test the other variants. This will be an interesting study in the context of more in-depth investigation of the protein engineering approach used but was deemed unnecessary for the purposes of this work.

- The term flux valve seems a bit misleading. Maybe this is more appropriately a flux flow restrictor? The work described seems to be flux tuning. A valve can be opened and closed and adjusted but there is only one state for the “valve” in each design.

Response:

We thank Reviewer 3 for this suggestion. As shown in Fig. 3a, the GGPPS can regulate the FPP flux distribution into carotenoid synthesis pathway and sterol synthesis pathway respectively. When using a lower-activity GGPPS, more flux is directed into the sterol synthesis pathway (See below response). By contrast, higher-activity GGPPS direct more substrate flux into carotenoid, such as lycopene synthesis in this study. Similar metabolic valve has also been reported (Ignea et al., 2019). Thus, we think the valve description is appropriate in this work.

- Lines 98 and 99: The manuscript talks about introducing an additional copy of GGPPS from *X. dendrorhous*. Why was *X. dendrorhous* GGPPS chosen over *M. circinelloides* GGPPS even though the strain harboring *M. circinelloides* β -carotene synthetic genes produced 4.12-fold more β -carotene? *M. circinelloides* GGPPS may have expressed better and resulted in even higher titers.

Response:

We only selected the GGPP synthases that have been previously reported in the metabolic engineering of microbes, especially yeast, to ensure that proper expression *in vivo* was not an issue. To our knowledge, we haven't found any reports that used the *M. circinelloides* GGPPS in yeast engineering. Furthermore, we believe that the *X. dendrorhous* GGPPS already provided sufficient

activity to achieve a pathway with high flux. Thus, we did not opt to test the GGPPS from *M. circinelloides*.

- Lines 168 and 169: The manuscript mentions the use of lower activity GGPPS for achieving high β -carotene titers. Decreasing GGPPS activity may result in the accumulation of FPP. FPP accumulation may result in a competing pathway to produce ergosterol. Was any kind of analytical quantification performed to identify FPP/ergosterol?

Response:

FPP content was measured in our experiments, and we found no statistically significant differences when using the lower-activity GGPPS compared to the original one (See below Fig. a). On the other hand, the strains harboring the lower-activity GGPPS did indeed accumulate more squalene (the precursor of ergosterol synthesis) than that of the original strain (See below Fig. b), confirming the reviewer's concern. Nevertheless, the benefits of restricting flux through GGPPS (and diverting it to ergosterol) to mitigate the substrate inhibition effect outweigh the loss of resources to a competing pathway.

- It would be better to add the IUP pathway shown in SF1 in the main paper figures.

Response:

We thank Reviewer 3 for this suggestion. IUP has been added to Fig.1 in the revised manuscript.

- Is the improvement of titer upon addition of the IUP pathway (SF12) statistically significant? It does not appear so by eye.

Response:

Yes, it is. The statistical difference (P value) analyzed by the Student's t-test was 0.0101. $P < 0.05$ was considered to be statistically significant.

- Please provide the data to support how 30 mM isoprenol and 10 mM prenol was found to be optimal.

Response:

Due to their toxicity, isoprenol and prenol can inhibit cell growth. Based on the cell growth curve after titrating the concentrations of isoprenol and prenol, we selected 30 mM isoprenol or 10 mM prenol as suitable levels (Supplementary Fig. 10). The word “optimal” has been replaced with “suitable” in the revised manuscript.

Reviewers' Comments:

Reviewer #1:

Remarks to the Author:

Following modifications made by the authors, manuscript NCOMMS-21-23818A can be accepted for publication after one modification (see below). The referee # 1 apologizes about one of his requests which was not sufficiently precise. The request was to show the developed (structural) formulas of the following molecules: IPP, FPP, GGPP, phytoene, lycopene, beta-carotene and not their crude formulas. Thanks to the authors to accede this request.

Reviewer #2:

Remarks to the Author:

I appreciate the response from the authors. The works are great and wonderful. But, I raised some points to be considered from the viewpoint of readers. I just hope the manuscript can be improved.

1) Title can be changed to be clear: "Removal of substrate inhibition enables high carotenoid productivity ~" to "Removal of lycopene substrate inhibition enables high carotenoid productivity ~".

2) Do the authors think that the lycopene substrate inhibition issue can be generalized to other lycopene cyclases than CarRP? The readers might be confused about it.

3) I wonder the truncated CarR has the mutation of Y27R. There is no clarification of it in the response letter. The lycopene inhibition test can be carried out with both the truncated CarR and the truncated CarR(Y27R).

4) It might be incorrect the authors' saying of "It is also possible that the concentration of lycopene was not measured or reported in prior work as it is a side product.". The researcher of this field easily recognized the formation of lycopene by its red color. The carotenoids production is generally analyzed by HPLC. Once the single carotenoid formation is confirmed, then it can be analyzed by UV spectrophotometer.

Reviewer #3:

Remarks to the Author:

The authors did a nice job of addressing most of the reviewer concerns; however, this reviewer disagrees with two of the responses.

First, the issue of measuring protein concentration of lycopene synthase raised by Reviewer #2. The logic supporting that lack of expression is sound but not definitive so the question still remains. While it is true that trying to overexpress lycopene cyclase by increasing its gene copy number and using a stronger promoter MAY increase protein levels, it also may not. There are numerous examples where resource allocation burden results in no improvements in protein level, despite higher mRNA (which was not even shown) and multigene copies. Therefore, the protein levels should in the best case be measured. At the very least, a thorough discussion with the author's logic, as stated in the response, should be included in the manuscript to allow the reader to evaluate the claim.

Second, the use of the term "flux valve" as raised by Reviewer #3 was not satisfactorily addressed. To use an illustrative analogy, your shower has a valve controlling the proportion of hot water coming out. If one's shower were built the way this flux valve was, you'd have to replumb a newly sized valve every time you showered or wanted a different temperature. I don't think avoiding the inaccuracy of the term flux valve (despite its past inaccurate use) diminishes the work in any way – and in fact strengthens it.

The response also noted the authors were not aware of *M. circinelloides* expression in yeast, so the following references are provided:

<https://microbialcellfactories.biomedcentral.com/articles/10.1186/s12934-018-0984-x>
<https://microbialcellfactories.biomedcentral.com/articles/10.1186/s12934-020-01309-0>

Response to Reviewers' comments

Reviewer #1

Following modifications made by the authors, manuscript NCOMMS-21-23818A can be accepted for publication after one modification (see below). The referee # 1 apologizes about one of his requests which was not sufficiently precise. The request was to show the developed (structural) formulas of the following molecules: IPP, FPP, GGPP, phytoene, lycopene, beta-carotene and not their crude formulas. Thanks to the authors to accede this request.

Response:

Structural formulas have been added to the revised manuscript.

Reviewer #2

I appreciate the response from the authors. The works are great and wonderful. But, I raised some points to be considered from the viewpoint of readers. I just hope the manuscript can be improved.

1) Title can be changed to be clear: "Removal of substrate inhibition enables high carotenoid productivity ~" to "Removal of lycopene substrate inhibition enables high carotenoid productivity ~".

Response:

We thank Reviewer 2 for this suggestion and have changed the title in the revised manuscript.

2) Do the authors think that the lycopene substrate inhibition issue can be generalized to other lycopene cyclases than CarRP? The readers might be confused about it.

Response:

The concern regarding generalization of the substrate inhibition discovered in lycopene cyclase of CarRP is an interesting one, but beyond the scope of this study. Our work can be the inspiration for other researchers to attempt similar methods in easing similar type of substrate inhibition.

3) I wonder the truncated CarR has the mutation of Y27R. There is no clarification of it in the response letter. The lycopene inhibition test can be carried out with both the truncated CarR and the truncated CarR(Y27R).

Response:

We apologize for this lack of clarity in our description. The truncated CarR in last response letter has the mutation of Y27R, named Truncated CarR (Y27R). The truncated CarR (wild type) was also used in carrying out inhibition assays (see below). These results below showed that substrate inhibition exists within CarRP, even its truncated CarR version, and mutation of Y27R can relieve this inhibition, regardless of the whole type CarRP (Y27R) or truncated CarR (Y27R).

4) It might be incorrect the authors' saying of "It is also possible that the concentration of lycopene was not measured or reported in prior work as it is a side product.". The researcher of this field easily recognized the formation of lycopene by its red color. The carotenoids production is generally analyzed by HPLC. Once the single carotenoid formation is confirmed, then it can be analyzed by UV spectrophotometer.

Response:

We agree with the Reviewer's viewpoint and have removed this sentence from the text.

Reviewer #3

The authors did a nice job of addressing most of the reviewer concerns; however, this reviewer disagrees with two of the responses.

First, the issue of measuring protein concentration of lycopene synthase raised by Reviewer #2. The logic supporting that lack of expression is sound but not definitive so the question still remains. While it is true that trying to overexpress lycopene cyclase by increasing its gene copy number and using a stronger promoter MAY increase protein levels, it also may not. There are numerous examples where resource allocation burden results in no improvements in protein level, despite

higher mRNA (which was not even shown) and multigene copies. Therefore, the protein levels should in the best case be measured. At the very least, a thorough discussion with the author's logic, as stated in the response, should be included in the manuscript to allow the reader to evaluate the claim.

Response:

Our data shows that before removing substrate inhibition, increasing gene copy number did not improve β -carotene production and the issue of lycopene accumulation remained (Supplementary Fig. 3b). On the contrary, after removing substrate inhibition, increasing gene copy number significantly improved β -carotene production as well as decreased lycopene level (Fig. 3e). Therefore, combination of these results suggests that there was no resource allocation burden that affected protein expression and protein level. In addition, increasing gene copy number also resulted in higher mRNA level (see below). These results suggest that the protein level of lycopene cyclase was not the limitation in our study. Both our *in vitro* (Fig. 1d) and *in vivo* (Fig. 3b and c) results demonstrate that substrate inhibition exists within CarRP, which was the main bottleneck resulting in lycopene accumulation. These data and new discussion have been added to the revised manuscript.

Second, the use of the term “flux valve” as raised by Reviewer #3 was not satisfactorily addressed. To use an illustrative analogy, your shower has a valve controlling the proportion of hot water coming out. If one's shower were built the way this flux valve was, you'd have to replumb a newly sized valve every time you showered or wanted a different temperature. I don't think avoiding the inaccuracy of the term flux valve (despite its past inaccurate use) diminishes the work in any way – and in fact strengthens it.

Response:

We thank the Reviewer for this illustration. His/her suggestion was accepted and we changed “flux valve” to “flux flow restrictor” in the revised manuscript.

The response also noted the authors were not aware of *M. circinelloides* expression in yeast, so the following references are provided:

<https://microbialcellfactories.biomedcentral.com/articles/10.1186/s12934-018-0984-x>

<https://microbialcellfactories.biomedcentral.com/articles/10.1186/s12934-020-01309-0>

Response:

We thank Reviewer 3 for kindly providing references.

Reviewers' Comments:

Reviewer #1:

Remarks to the Author:

Referee 1 considers that the revised manuscript can be published as is.

Reviewer #2:

Remarks to the Author:

None.

Reviewer #3:

Remarks to the Author:

This reviewer thanks the authors for their continued thoughtful responses to reviewer comments and questions. The manuscript can now be published in this reviewer's opinion.

Response to Reviewers' comments

Reviewer #1

Referee 1 considers that the revised manuscript can be published as is.

Response:

We appreciate your positive and constructive comments.

Reviewer #2

None.

Response:

We appreciate your positive and constructive comments.

Reviewer #3

This reviewer thanks the authors for their continued thoughtful responses to reviewer comments and questions. The manuscript can now be published in this reviewer's opinion.

Response:

We appreciate your positive and constructive comments.